# Design of Dual-Band Terahertz Perfect Metamaterial Absorber Based on Circuit Theory

**DOI:** 10.3390/molecules25184104

**Published:** 2020-09-08

**Authors:** Zhongmin Liu, Liang Guo, Qingmao Zhang

**Affiliations:** 1Guangzhou Key Laboratory for Special Fiber Photonic Devices and Applications, South China Normal University, Guangzhou 510006, China; Liuzm@m.scnu.edu.cn (Z.L.); guoliangchn@163.com (L.G.); 2Guangdong Provincial Key Laboratory of Nanophotonic Functional Materials and Devices, South China Normal University, Guangzhou 510006, China

**Keywords:** dual-band absorber, graphene metasurface, circuit model

## Abstract

We present a novel strategy for designing a dual-band absorber based on graphene metasurface for terahertz frequencies. The absorber consists of a two-dimensional array of patches deposited on a metal-backed dielectric layer. Using an analytical circuit model, we obtain closed-form relatinos for the geometrical parameters of the absorber and the properties of the applied materials to achieve the dual-band absorber. Two absorption bands with perfect absorption at the preset frequencies of 0.5 and 1.5 THz are achieved. The results obtained by the analytical circuit model are compared to the simulations carried out by full-wave electromagnetic field analysis. The agreement between results is very good. We demonstrate that the graphene absorber remains as the dual band for a wide range of the chemical potential. Furthermore, the recommended dual band absorber is insensitive in terms of polarization and remain within various incident angles.

## 1. Introduction

Metamaterials are artificially-made media with uniquely engineered electric permittivities and magnetic permeabilities [1,2] and enabled design of thin electromagnetic energy absorbers [3]. Metamaterial-based absorbers have attracted considerable attention in the terahertz (THz) region due to their promising applications in imaging [4], biosensing [5,6], optical control [7], etc. Recently, the realization of dual-band terahertz absorbers based on metamaterials has been investigated [8,9,10,11,12,13,14]. Such dual-band absorbers can be utilized in dual-band transceiver systems and enhance energy absorption for imaging and biochemical sensing. In these structure, various mechanisms of dual-band absorption have been recommended by a combination of multiple resonators with a hybrid structure metamaterial layer or stacked structure metamaterial layers. However, these works require not only the structure of the complex, but also the corresponding tedious design and fabrication process. Therefore, designing simple structures with an efficient method is of great importance [15,16,17,18,19].

To realize metamaterial-based absorbers in the low THz region, one can use the graphene patterned arrays. Graphene, consisting of a two-dimensional layer of carbon atoms arranged in a honeycomb lattice, has recently been demonstrated as one of the best materials for many applications in electrical and photonic devices due to its exotic properties [20,21,22,23], such as high optical transparency [24], ultrafast electronic transport [25], and controllable electric conductivity, which can be tuned by changing the chemical potential [26]. Furthermore, graphene has the ability to supporting surface plasmon polaritons in the terahertz and infrared ranges [27]. In addition, in comparison to surface plasmon polaritons in noble metals, graphene plasmon polaritons display greater ability to achieve high local field, subdiffraction confinement, and low optical losses [28,29].

In this paper, a simple structure, composed of a period array of graphene patches deposited on a dielectric spacer terminated by a metallic back reflector, is used to design a dual-band THz absorber. We propose a circuit model to extract the geometrical parameters of the absorber and the properties of the material in the closed-form. Furthermore, we present a simple and effective approach for achieving perfect absorption at two preset frequencies. The input admittance of the absorber at the two frequencies is designed to match the admittance of free-space. Moreover, the imaginary part of the input admittance at the central frequency is set by applying an additional condition to be zero. To validate the approach, dual-band perfect THz absorbers have been investigated and designed. Then, full-wave simulations are used to confirm the design method. Excellent agreements have been achieved. It should be noted that the recommended absorber is polarization insensitive for normal incident electromagnetic waves owing to its symmetric structure. Besides, both dual-band and the high absorption effectiveness are affected slightly even at high angles of incidence for both TE and TMpolarizations.

The paper is organized as follows. The equivalent circuit model of the proposed device is introduced, and then a design scheme leading to an dual-band absorber is proposed in Section 2. Numerical validation of the accuracy and effectiveness of the model as well as discussions are presented in Section 3. Finally, conclusions are drawn in Section 4.

## 2. Design Method

The geometry of the proposed absorber is schematically shown in Figure 1, in which the top layer of periodic array of graphene patches with period *D* and gap *g* between the patches and the bottom layer of a reflective golden plate with finite conductivity of $4\times 10^{7}$ S/m are separated by a dielectric spacer with relative permittivity $\varepsilon_{d}$ and thickness *d*. It is worth mentioning that in the fabrication process, the Si is used to support the graphene patches. The Au film is deposited on the Si bottom by gilding. Next, a continuous and uniform atomic monolayer of graphene can be grown by the chemical vapor deposition method, then patterned into closely packed square patch arrays using e-beam lithography.

Based on transmission line theory, the whole structure can be modeled as circuit elements [30]. The equivalent circuit of the proposed structure at normal incidence for TM polarization (electric field along the *x* direction) is demonstrated in Figure 2. The graphene patch array can be modeled by a shunt admittance $Y_{g}$. The homogeneous regions namely the free space, the dielectric spacer and the Au backplate are modeled as transmission lines with the admittances $Y_{0}$, $Y_{d}$ and $Y_{Au}$, respectively. It should be pointed out that the metallic back reflector which is suppressing the transmittance can be approximately considered as a short circuit.

An accurate circuit model has been proposed by Barzegar-Parizi et al. for a graphene patch array illuminated by a normally incident plane wave [31]. The admittance $Y_{g}$ in the model represents an infinite number of parallel R-L-C circuits, each corresponding to a mode of the structure.
(1)$$Y_{g}=\sum_{n=1(odd)}^{\infty }{\left(R_{n}+j\omega L_{n}+\frac{1}{j\omega C_{n}}\right)}^{-1}$$
in which ω=2πf shows the angular frequency and the values of $R_{n}$, $L_{n}$, and $C_{n}$ are calculated as
(2)$$R_{n}=\frac{D^{2}K_{n}}{S_{n}^{2}}Re\left\{\sigma_{g}^{-1}\right\},L_{n}=\frac{D^{2}K_{n}}{S_{n}^{2}}\frac{Im\{\sigma_{g}^{-1}\}}{\omega },C_{n}=\frac{S_{n}^{2}}{D^{2}K_{n}}\frac{2\varepsilon_{eff}}{q_{n}}$$
where $S_{n}=\int_{S}\xi_{n}\left(x,y\right)dS$, $K_{n}=\int_{S}{\left|\xi_{n}\left(x,y\right)\right|}^{2}dS$, in which *S* is the surface of a patch, $\xi_{n}\left(x,y\right)$, and $q_{n}$ are the n’th normalized eigenfunction and eigenvalue corresponding to n’th the resonant modes excited by the array of the patches. $\varepsilon_{eff}=\varepsilon_{0}\left(1+\varepsilon_{d}\right)/2$ is the average permittivity of the mediums surrounding the graphene patches, $\sigma_{g}$ is the surface conductivity of the graphene.

It is known that graphene’s conductivity is derived using the Kubo formula, which is described with interband and intraband contributions [32,33]. For the THz frequencies, the photon energy $\hbar \omega <<2\mu_{c}$ ($\mu_{c}$ is the chemical potential), the interband part is negligible comparing to the intraband [33,34]. Therefore, graphene can be described by the Drude-like surface conductivity in the THz range:(3)$$\sigma_{g}=\frac{\sigma_{0}}{1+j\omega \tau }$$
where
(4)$$\sigma_{0}=\frac{e^{2}k_{B}T\tau }{\pi \hbar^{2}}\left\{\frac{\mu_{c}}{k_{B}T}+2ln\left(e^{-\mu_{c}/k_{B}T}+1\right)\right\}$$
where *e* is the electron charge, $k_{B}$ is Boltzmann’s constant, and *ℏ* is the reduced Planck’s constant. *T* is the temperature (fixed to 300 K) and τ is the relaxation time of graphene.

Designing the proposed absorber near the first resonance frequency of graphene patches, we can only consider the first mode and neglect the effect of higher-order modes. To avoid numerical calculation of the eigenvalues in the previous capacitance expression, we propose to replace the expression with the analytical capacitance admittance of metal patches proposed in [35]. Therefore, the surface admittance $Y_{g}$ of the graphene patches can be rewritten as follows,
(5)$$Y_{g}={\left(R+j\omega L+\frac{1}{j\omega C}\right)}^{-1}$$
where
(6)$$R=\frac{1.13D^{2}}{{\left(D-g\right)}^{2}}Re\left\{\sigma_{g}^{-1}\right\},L=\frac{1.13D^{2}}{{\left(D-g\right)}^{2}}\frac{Im\{\sigma_{g}^{-1}\}}{\omega },C=\frac{2\varepsilon_{eff}}{\pi }Dln\left\{csc\left(\frac{\pi g}{2D}\right)\right\}$$

In the equivalent circuit model of Figure 2. $\beta =nk_{0}$ and $Y_{d}=nY_{0}$ are the propagation constant and the admittance of transmission line matching the dielectric spacer, respectively, where $k_{0}=\omega /c$ (*c* is the speed of light) is the wavenumber of free space and the free-space admittance $Y_{0}=1/\left(120\pi \right)$. Therefore, the total input admittance of the device is obtained as
(7)$$Y_{in}=Y_{g}-jY_{d}cot\left(\beta d\right)$$

Suppose perfect absorption can be achieved at two frequencies $\omega_{1}$ and $\omega_{2}$ ($\omega_{1}$ < $\omega_{2}$). Accordingly, the conditions for this to happen can be summarized as
(8a)$$Im\left(Y_{in}\right){|}_{\omega =\omega_{1}}=0$$
(8b)$$Re\left(Y_{in}\right){|}_{\omega =\omega_{1}}=Y_{0}$$
and
(9a)$$Im\left(Y_{in}\right){|}_{\omega =\omega_{2}}=0$$
(9b)$$Re\left(Y_{in}\right){|}_{\omega =\omega_{2}}=Y_{0}$$

We consider a central frequency defined as $\omega_{0}=\left(\omega_{1}+\omega_{2}\right)/2$ and apply an additional condition on the structure by setting ${\beta |}_{\omega =\omega_{0}}d=\pi /2$. The condition leads to $LC=\omega_{1}\omega_{2}$. Thus, the resonance frequency of the graphene array can be realized between dual band. Then, the thickness of the dielectric slab is obtained as
(10)$$d=\frac{\pi c}{2\sqrt{\varepsilon_{d}}\omega_{0}}$$

Then, the admittance of the graphene array is seen as the input admittance of the structure, and it can be tuned to be matched to the free-space admittance near its resonant frequency [36]. After straightforward mathematical manipulations, from Equations ([Disp-formula FD8a-molecules-25-04104]) and ([Disp-formula FD9a-molecules-25-04104]) we obtain
(11)$$\tau =\frac{\sqrt{\varepsilon_{d}}cot\left(\frac{\pi \omega_{1}}{\omega_{2}+\omega_{1}}\right)}{\omega_{2}-\omega_{1}}$$

Choosing the two frequencies $\omega_{1}$ and $\omega_{2}$, one can obtain the relaxation time of graphene by Equation (Equation 11) for given values of *n*. The relaxation time can be tuned by the chemical potential through the relation $\tau =\left(\mu \mu_{c}\right)/\left(ev_{f}^{2}\right)$, in which $v_{f}=10^{6}$ m/s is the Fermi velocity and μ is the electron mobility ranging from approximately 0.03 $m^{2}$/Vs to 6 $m^{2}$/Vs, depending on the fabrication process [37,38,39,40,41]. Therefore, by computing the relaxation time, the chemical potential of graphene is computed by
(12)$$\mu_{c}=\frac{\tau ev_{f}^{2}}{\mu }$$

At the end, the geometrical parameters of the structure are computed by Equations (Equation 6)–(Equation 10) as
(13)$$R=\frac{1}{Y_{0}\left(1+\tau^{2}{\left(\omega_{2}-\omega_{1}\right)}^{2}\right)}\Rightarrow \frac{g}{D}=1-\sqrt{\frac{1.13Y_{0}\left(1+\tau^{2}{\left(\omega_{2}-\omega_{1}\right)}^{2}\right)}{\sigma_{0}}}$$
(14)$$LC=\frac{1}{\omega_{1}\omega_{2}}\Rightarrow D=\frac{\pi \sigma_{0}}{2.26\tau \varepsilon_{eff}\omega_{1}\omega_{2}}{\left(1-\frac{g}{D}\right)}^{2}ln^{-1}\left\{csc\left(\frac{\pi g}{2D}\right)\right\}$$

## 3. Numerical Results and Discussion

With the aid of the analytical equations presented in Section 2, dual-band perfect graphene absorbers are designed in this part. The performances of the designed absorbers are verified numerically using full-wave simulations (Lumerical Nanophotonic FDTD simulation software). Suppose the relative permittivity of the dielectric spacer is $\varepsilon_{d}=11.9$ corresponding to Si and the frequencies of the first band and second band are $f_{1}=0.5$ THz and $f_{2}=1.5$ THz, respectively. Therefore, the central frequency is calculated as $f_{0}=1$ THz. The value of the relaxation time is given as $\tau =5.49\times 10^{-13}$ s from Equation (Equation 11). The electron mobility is assumed to be μ=0.75
$m^{2}$/Vs. Thus, using Equation (Equation 12), we have $\mu_{c}=0.732$ eV. The thickness of the dielectric spacer is calculated by Equation (Equation 10) as d=21.73 μm. Finally, from Equations (Equation 13) and (Equation 14), the patch width and the period are extracted as g=2.93 μm and D=30.54 μm, respectively. With these procedures, a dual-band perfect absorber is designed.

The absorption spectra of the structure are plotted in Figure 3a, calculated from the equivalent circuit model approach and compared with that obtained from FDTD simulation. An excellent agreement is observed between the two results, demonstrating the equivalent circuit model is accurate and effective. Figure 3b reveals the real and imaginary parts of the normalized input admittance as a function of frequency for the absorber. As it can be observed, the imaginary part of the normalized input admittance is zero at the two frequencies of $f_{1}$ and $f_{2}$, whereas the real part matches free space admittance perfectly. Therefore, the prepared conditions is fulfilled for impedance matching at two bands and a perfect dual-band absorption is achieved. All key material and geometrical parameters of the absorber can be calculated directly from the formulas proposed according to design requirements. The equivalent circuit model approach speeds up the absorber design and analysis.

It should be pointed out that there are slight disagreements between the circuit model and FDTD simulation. These slight disagreements result from neglecting the effect of higher-order modes of graphene patches in the circuit model shown in Figure 2, which was used in the design process. However, the effect of the higher-order modes has been included in the simulations.

Now, the tunability of the proposed device is investigated by changing the chemical potential of graphene. Figure 4 shows the absorption spectra for the chemical potential ranging between 0.4 eV and 0.6 eV. It can be observed that the first resonant frequency is slightly shifted to a higher frequency region with increasing the chemical potential, whereas the absorption changes considerably. However, the location of the second resonant band moves to higher frequencies significantly with increasing the chemical potential while its absorption peak changes slightly.

The polarization insensitive performance for the proposed absorber is important in practical applications. We plot the absorption spectra as a function of polarization angle and frequency in Figure 5. Due to the symmetrical design of the structure, it is clearly seen that the absorption spectra of the proposed dual-band absorber is polarization insensitive.

We have also investigated the omnidirectional characteristic for the proposed absorber. Figure 6 plots the absorption spectra of the absorber as function of frequency and incident angle for TE and TM polarization, respectively. The two absorption peaks can be observed within a wide range of incident angles for both TE and TM polarizations. Moreover, at the incident angle below 60°, a substantial overlap can be observed between the absorption spectra for both polarizations. Therefore, the proposed device can also be verified to be a non-polarization-dependent absorber within this range.

## 4. Conclusions

In summary, a dual-band perfect absorber based on graphene patches has been designed in the low terahertz regime by using impedance matching concept. First, we assigned an equivalent circuit to the absorber. We then adjusted the real part of the input admittance of the absorber to be close to the free space admittance and the imaginary part of the input admittance to be zero at two frequencies. In addition, the dielectric slab is considered as a quarter wavelength transmission line at the center of the two frequencies. This method resulted in closed-form relations for the geometry of the structure and the characteristics of the applied material. According to the results of simulation and analytical circuit model, the recommended absorber can operate with perfect absorption at 0.5 THz and 0.15 THz. Further, the device are insensitive to polarization and omnidirectional.

## Figures and Tables

**Figure 1 molecules-25-04104-f001:**
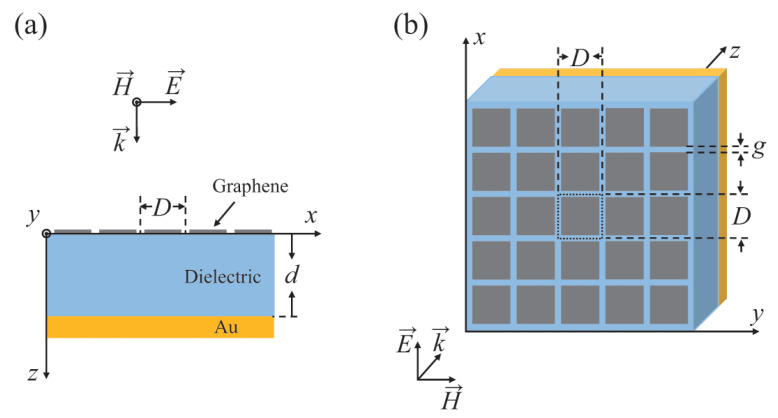
Schematic diagram of the proposed absorber: (**a**) cross section view and (**b**) 3D view.

**Figure 2 molecules-25-04104-f002:**
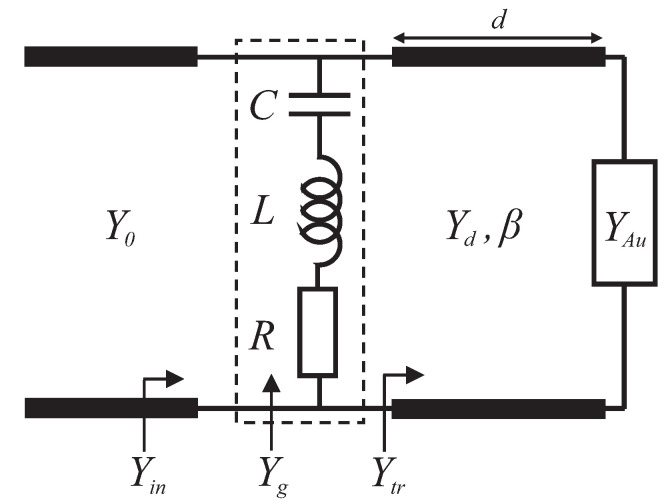
Equivalent circuit model for the graphene metasurface-based absorber.

**Figure 3 molecules-25-04104-f003:**
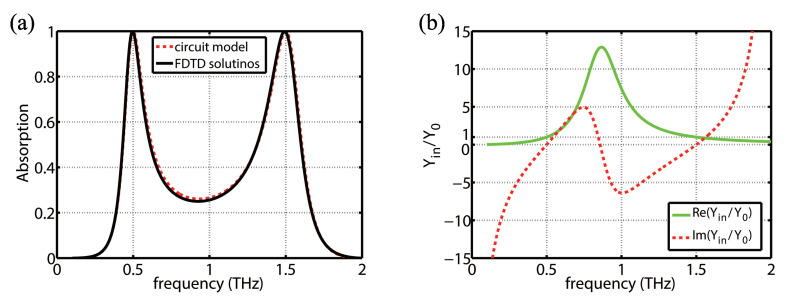
(**a**) The absorption spectrum of the designed dual-band absorber with the two frequencies at 0.5 THz and 1.5 THz calculated by the circuit model and FDTD simulation. (**b**) The real and imaginary parts of the normalized input admittance of the designed absorber.

**Figure 4 molecules-25-04104-f004:**
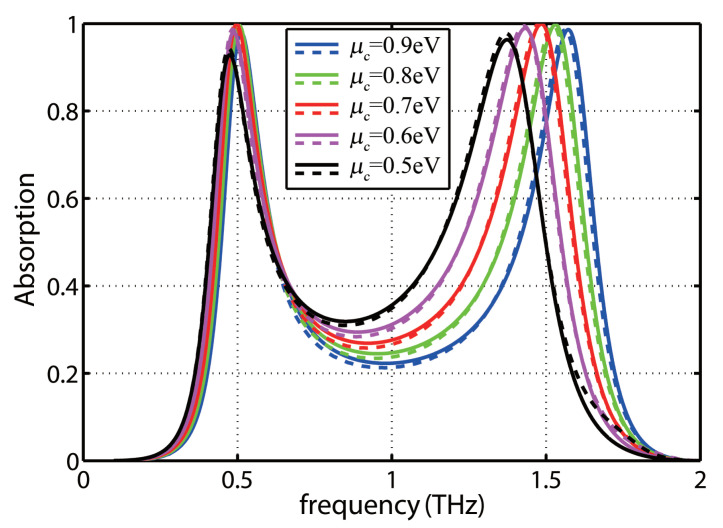
The absorption spectrum of the graphene-based proposed absorber for different chemical potential of graphene.

**Figure 5 molecules-25-04104-f005:**
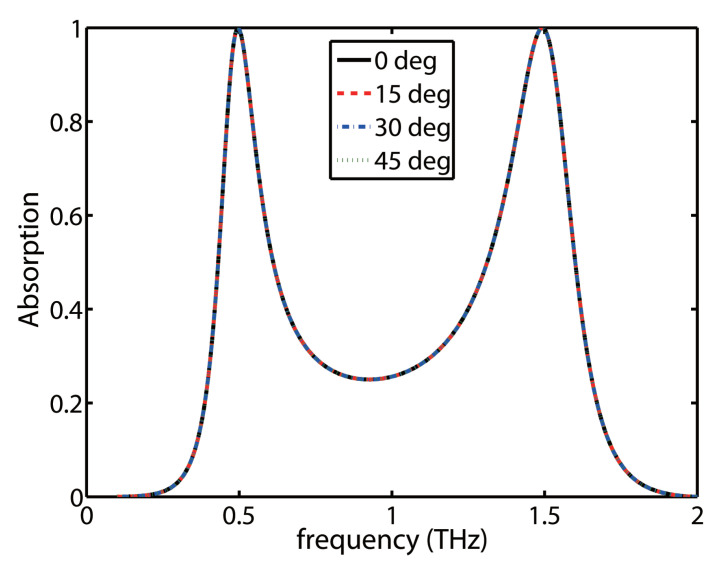
The absorption spectrum of the proposed absorber as a function of polarization angle and frequency.

**Figure 6 molecules-25-04104-f006:**
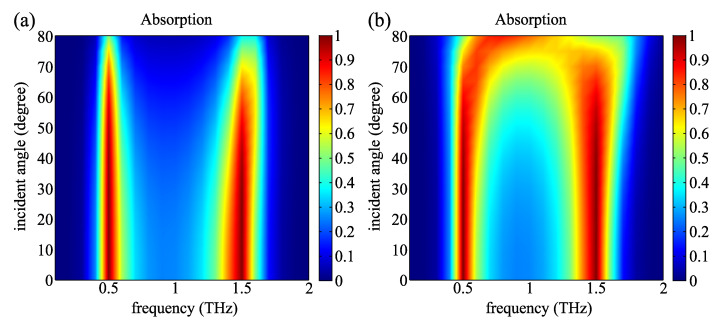
The absorption spectrum of the proposed absorber as a function of incident angle and frequency for (**a**) TM and (**b**) TE polarizations.

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
