# Peer review of "Design of Dual-Band Terahertz Perfect Metamaterial Absorber Based on Circuit Theory"

_molecules, 2020, doi:10.3390/molecules25184104_

Round 1
Reviewer 1 Report
The authors proposed a design of perfect absorber using graphene metasurface. The authors should answer the following questions:
- The proposed perfect absorber is narrowband, what is the advantage of it compared to other narrowband perfect absorber?
- In the paper, the author assumed the conductivity of gold is a constant. In the reality, the conductivity changes with frequency. Will the results change a lot considering the varying conductivity?
- Although it is a theoretical work, the author should discuss the potential way to fabricate this perfect absorber.
Reviewer 2 Report
- The authors should clarify the following sentence (see page 4):
"To better material selection and geometry optimization, we consider a central frequency defined as ω0 = (ω1 + ω2)/2 and apply an additional condition on the structure by setting β|ω=ω0d = π/2"
The authors should explain why they apply this additional condition. What is its physical meaning?
- In order to demonstrate the polarization-insensitive behavior of the proposed metamaterial absorber, the absorption spectrum should be evaluated for different values of the azimuthal angle (ranging from 0° up to 360°) of the impinging plane wave. Please include this information.
- Figure 3 is incomplete; I cannot see part (b). Please, add Fig. 3(b).
Round 2
Reviewer 2 Report
The English language is not always adequate, so I suggest to find a fluent English speaker to revise the whole manuscript.